# ODC: Orthogonal Drift Correction for Improved Text-to-Image Semantic Alignment at Inference

## Abstract

Text-to-image models have achieved remarkable success in generating high-quality images from textual descriptions. However, they often struggle with "semantic drift," where the generated output fails to precisely align with complex or nuanced text prompts. While recent approaches have attempted to address semantic errors regarding attribute binding or object presence, there remains a gap for a more holistic method that addresses these issues by directly refining the text embeddings of the initial user prompt. In this work, we introduce Orthogonal Drift Correction (ODC), an inference-time guidance method designed to mitigate semantic drift without requiring model retraining or additional user inputs. ODC guides image generation through a two-stage process. In the first stage, it identifies the semantic drift by evaluating the initially generated image against the user prompt in a shared vision-language embedding space. It then isolates the component of this drift vector that is orthogonal to the prompt's direction and translates it back into text via a vocabulary-based surrogate mechanism. In the second stage, it produces refined text conditioning for a second generation pass by feeding both the initial text embedding and the re-embedded drift representation into an adaptive rank-reduced concept removal module. Our experiments demonstrate the effectiveness of ODC in enhancing prompt-image alignment, yielding images that more accurately reflect detailed compositional instructions. As a plug-and-play module, ODC offers a practical and efficient method for improving the reliability of state-of-the-art text-to-image models.

## 1 Introduction

Modern text-to-image (T2I) models have opened new creative possibilities, allowing people to generate compelling images that would have been impossible to generate just a few years ago. Models such as Stable Diffusion (Rombach et al., 2022a), Imagen (Saharia et al., 2022), and DALL-E 2 (Ramesh et al., 2022) have demonstrated an unprecedented ability to synthesize visually compelling and artistic images from natural language descriptions, unlocking wide-ranging applications across numerous domains.

However, despite these profound capabilities, current state-of-the-art T2I models frequently fall short in generating images that precisely align with the full semantic meaning and composition of their input text prompts. This lack of precise semantic alignment, a phenomenon we term "semantic drift", pertains to the frequent failure of these models to precisely adhere to user prompts. For instance, they often misinterpret complex compositional instructions, leading to incorrect attribute binding (e.g., wrong material on objects), catastrophic neglect (Chefer et al., 2023) (omitting subjects from the prompt), or a failure to adhere to specified spatial relationships. This alignment gap forces users into frustrating trial-and-error prompt engineering, a practice that underscores the limitations of such models in accurately interpreting semantic intent.

This difficulty largely stems from how T2I models process textual information. State-of-the-art T2I diffusion models employ a U-Net architecture with cross-attention layers (Rombach et al., 2022a), which are crucial for fusing visual and textual features. While these layers encode rich semantic relations, the interaction can sometimes skew feature representations, causing semantic misalignment between the input and output. A variety of approaches have been proposed to address this problem. Fine-tuning methods like DreamBooth

(Ruiz et al., 2023) and CoMat (Jiang et al., 2024), personalize models for specific subjects but can be computationally demanding. Architectural expansions like ControlNet (Zhang et al., 2023) and T2I-Adapter (Mou et al., 2024) provide explicit spatial control through auxiliary inputs but require additional inputs and specialized training. Alternatively, a growing class of methods focuses on training-free, inference-time manipulation of attention maps or latent representations. Techniques such as Attend-and-Excite (Chefer et al., 2023), CONFORM (Meral et al., 2024), and ToMe (Hu et al., 2024) directly intervene in the generation process to ensure robust semantic binding and alignment. While effective for specific tasks such as subject presence and attribute binding, they do not address the misalignment caused by imperfect initial text embeddings. This body of work reveals a clear gap for a method that can efficiently improve text-image alignment by directly correcting the text embedding to resolve complex compositional failures.

In this paper, we address this gap by introducing Orthogonal Drift Correction (ODC), a unique inference-time technique that directly corrects the text conditioning vector to prevent semantic drift. Our approach is based on the key insight that semantic drift originates from the components of the computed text embedding that are orthogonal to the prompt's true semantic direction. ODC guides the image generation process through a two-stage mechanism. First, it generates an initial candidate image. Then, it leverages the shared embedding space of a pre-trained vision-language model, to identify the semantic error vector between the generated image and the prompt. Crucially, instead of using the entire error, it isolates its orthogonal component via vector rejection. This component, which intuitively represents the "off-topic" deviation, is then removed from the initial text embedding. The resulting purified embedding guides a second generation pass, producing a final image with substantially higher prompt-image alignment.

Our contributions can be summarized as follows:

- We identify *orthogonal semantic drift* in text embeddings as a key source of prompt-image misalignment in text-to-image models, and propose a model-agnostic, training-free, inference-time method that corrects the text conditioning embeddings without modifying the diffusion process or requiring fine-tuning.

- We propose an adaptive rank-reduced concept removal strategy that refines the text conditioning by processing the initial text embedding alongside a re-embedded drift representation, obtained via a vocabulary-based surrogate mechanism.

- We introduce CoALBench-300 (**CO**mpositional **AL**ignment Benchmark), 300 synthetically generated prompts curated to rigorously probe semantic drift and prompt-image alignment. To support our method, we curate a large-scale vocabulary of over 300,000 words and phrases, compiled from multiple sources and refined through several preprocessing steps, to effectively capture and mitigate semantic drift.

## 2 Background & Related Work

We provide the necessary context for understanding our proposed method below.

**Prompt-Image Alignment.** Prompt-image alignment refers to the faithfulness of a generated image to the full semantic meaning and compositional structure of its input text prompt. While the goal of Text-to-Image (T2I) models is to translate natural language into visually aligned results, achieving precise alignment in terms of the spatial composition, complex layouts, poses, shapes, and forms described in a prompt remains a significant challenge (Huang et al., 2023; Saharia et al., 2022). This often necessitates numerous trial-and-error cycles from users to achieve their desired output. Our method contributes to improving prompt-image alignment without requiring these cycles.

**Text-to-Image Models.** The dominant architecture in modern T2I generation is the Latent Diffusion Model (LDM) (Rombach et al., 2022b). Unlike earlier diffusion models that operated in the high-dimensional pixel space (Ho et al., 2020), LDMs perform the computationally intensive denoising process in the compressed latent space of a pre-trained variational autoencoder (VAE). This significantly reduces computational overhead while maintaining high-fidelity output.

The core of an LDM's denoising process is a U-Net architecture (Ronneberger et al., 2015), which comprises a series of downsampling and upsampling blocks with residual connections. Interspersed within this network are self-attention layers, which capture global spatial dependencies, and crucially, cross-attention layers. These cross-attention layers are the primary mechanism through which textual guidance is integrated into the image generation process. They allow the model to attend to different parts of the text prompt at each denoising step, conditioning the visual features being generated. More advanced models like Stable Diffusion XL (SDXL) (Podell et al., 2023) enhance this architecture with a larger U-Net and a dual text encoder setup to achieve superior performance.

**Text Embeddings and Conditioning.** The journey from a text prompt to a guiding signal begins with tokenization, where the input string is converted into a sequence of discrete tokens. These tokens are then mapped to high-dimensional embedding vectors. This sequence of vectors is processed by a powerful text encoder, such as the CLIP text encoder (Radford et al., 2021) or a larger language model like T5 (Raffel et al., 2020), to produce the final conditioning embedding.

This text embedding conditions the denoising U-Net at multiple resolutions via the cross-attention mechanism. At each cross-attention layer, the intermediate visual features (the query) attend to the text embedding (the key and value). This process generates spatial attention maps that define a rich semantic relationship between image regions and prompt tokens, critically influencing the final image's composition. Our method, intervenes at the very beginning of this pipeline. It is plugged into the process by modifying the conditioning embedding *before* it is passed to the U-Net, thereby correcting the guidance signal at its source.

**Previous Attempts to Improve Alignment.** The challenge of prompt-image misalignment has inspired a wide variety of solutions. Prior work can be broadly categorized into three categories: prompt engineering, training-based methods, and inference-time guidance.

- **Prompt Engineering:** At the most fundamental level, users engage in manual prompt engineering, iteratively refining text to achieve desired results. More structured approaches include prompt weighting, which allows users to amplify or attenuate the semantic influence of specific words, and the use of negative prompts. Negative prompts modify the unconditional embedding used in classifier-free guidance to explicitly steer the model away from unwanted concepts (Ho & Salimans, 2022). The efficacy of these prompts depends heavily on the capacity of the text encoder to capture complex relationships. Saharia et al. (2022) demonstrated that scaling the frozen text encoder (e.g., T5-XXL) yields significantly larger improvements in alignment than scaling the diffusion model itself. Beyond manual refinement and architectural scaling, researchers have also explored optimizing the input space directly to represent specific concepts. Textual Inversion (Gal et al., 2022) optimizes new pseudo-words within the embedding space of a frozen model to capture unique subjects or styles, acting as a bridge between the user's intent and the model's vocabulary.

- **Training-Based Methods:** A powerful but resource-intensive approach involves fine-tuning the model. DreamBooth (Ruiz et al., 2023) fine-tunes the entire diffusion model to bind a unique identifier with a specific subject, allowing for the generation of that subject in novel contexts with high fidelity. To mitigate the high computational cost of full fine-tuning, Low-Rank Adaptation (LoRA) (Hu et al., 2022) freezes pre-trained weights and injects trainable rank-decomposition matrices, enabling efficient adaptation to new tasks or styles with minimal parameter updates. While these methods focus on personalization, other works target general alignment failures such as concept ignorance. CoMat (Jiang et al., 2024) introduces an end-to-end fine-tuning strategy that leverages an image-to-text captioning model to minimize the discrepancy between the prompt and the generated image's caption, aligning the model to revisit ignored concepts and correct attribute mismapping.

- **Inference-Time Guidance:** This category includes methods that, like ours, operate at inference time without modifying model weights. The foundational technique in this category is Classifier-Free Guidance (CFG) (Ho & Salimans, 2022), which mixes conditional and unconditional score estimates to trade off sample diversity for higher text alignment. Building on this, Chung et al. (2024) identified that standard CFG suffers from "off-manifold" issues that harm editability and alignment, proposing CFG++ to correct

the sampling trajectory to remain on the data manifold. Beyond sampling adjustments, several methods intervene specifically in the cross-attention layers or latent optimization during inference to correct semantic failures. Hertz et al. (2022) introduced Prompt-to-Prompt, which injects cross-attention maps from a source generation into a new generation to maintain structural layout while editing semantic content. To address "catastrophic neglect", Attend-and-Excite (Chefer et al., 2023) optimizes the latent at each timestep to ensure all subject tokens maximize their attention activations. Similarly, CONFORM (Meral et al., 2024) applies a contrastive objective during inference to segregate objects in attention maps while keeping attributes close to their subjects. Taking a different approach to binding, Hu et al. (2024) proposed Token Merging (ToMe), which merges relevant tokens into a composite token and iteratively updates this embedding during the early denoising steps using semantic binding and entropy losses. Finally, rather than optimizing the denoising path, InitNO (Guo et al., 2024) focuses on the initialization phase. It partitions the initial latent space into valid and invalid regions based on attention conflict scores and optimizes the initial noise vector to ensure it is predisposed toward semantically faithful generation before denoising begins.

Crucially, our work differs from existing inference-time approaches in its point of intervention. Whereas prior methods act by altering the internal dynamics of the U-Net denoising process, our approach is simpler and more direct: we refine the conditioning embeddings before diffusion begins. In contrast to training-based approaches, our method is entirely training-free, requiring no fine-tuning or additional optimization. Similarly, unlike attention-manipulation techniques, our intervention operates directly on the text embeddings, making it straightforward to implement and agnostic to the choice of U-Net architecture. By correcting semantic drift at its source, ODC provides a generalizable and principled way to correct semantic drift at its source, improving the alignment and reliability of pre-trained text-to-image models.

## 3 Orthogonal Drift Correction

Our proposed method, Orthogonal Drift Correction (ODC), is an inference-time guidance technique designed to enhance the semantic alignment of text-to-image models. The central hypothesis is that suboptimal prompt alignment, or *semantic drift*, stems from the components of the initial text embedding that are orthogonal to the prompt's primary semantic direction. ODC operates in a two-stage process. An initial image is generated and then evaluated against the prompt in a shared vision-language embedding space. The identified semantic error is then used to refine the initial text embedding for a second, more accurate generation pass.

### 3.1 Notations

Let $p_{\text{txt}}$ denote the input text prompt. The text-to-image model consists of two text encoders, a token-level encoder $\mathcal{E}_{\text{tok}}$ and a pooled encoder $\mathcal{E}_{\text{pool}}$, together with an image generator $\mathcal{G}$ (typically a U-Net implementing a reverse diffusion process). The standard generation process first encodes the text prompt as

$$\mathbf{E}_p = \mathcal{E}_{\text{tok}}(p_{\text{txt}}), \qquad \mathbf{e}_p = \mathcal{E}_{\text{pool}}(p_{\text{txt}}), \tag{1}$$

where $\mathbf{E}_p \in \mathbb{R}^{L \times H}$ is the sequence of per-token embeddings and $\mathbf{e}_p \in \mathbb{R}^H$ is a pooled sentence-level vector. The image generator then produces an initial image $I$ conditioned on both representations:

$$I = \mathcal{G}(\mathbf{E}_p, \mathbf{e}_p). \tag{2}$$

To evaluate and correct for semantic drift, we employ a pre-trained vision–language model (VLM), denoted $\mathcal{E}_{\text{VLM}}$, which embeds both text and images into a shared multimodal space.

### 3.2 The Orthogonal Drift Correction (ODC) Algorithm

The ODC algorithm consists of two main stages, comprising six sequential steps in total. These steps are described in detail below and summarized in Algorithm 1. An overview of the process is illustrated in Figure 1.

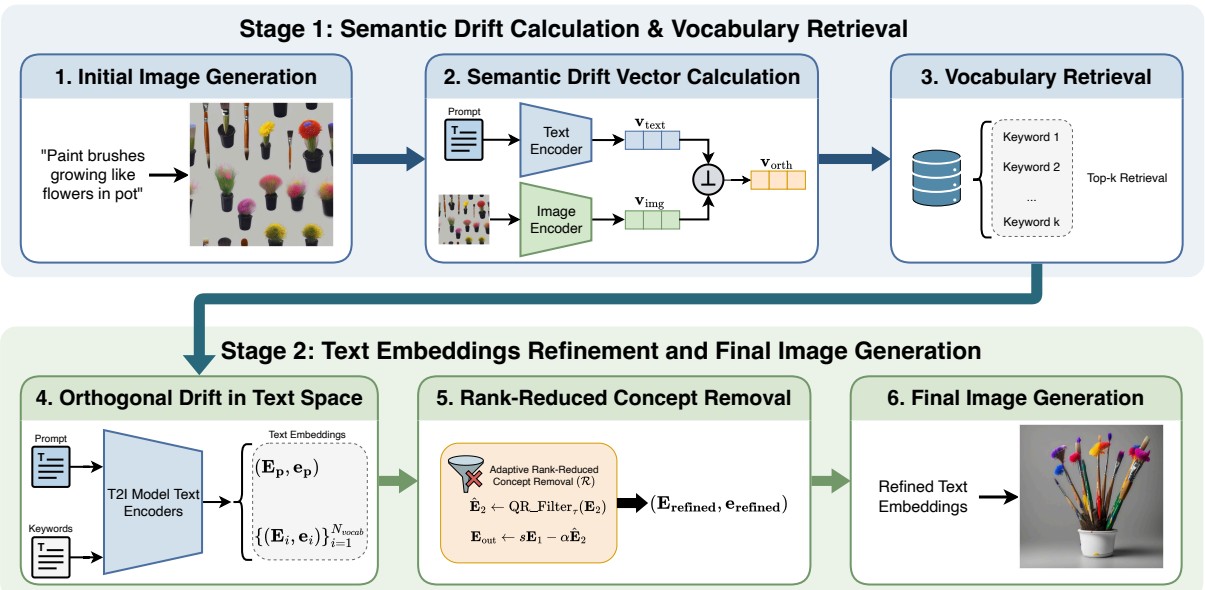

Figure 1: The workflow of Orthogonal Drift Correction (ODC). In the first stage, vocabulary items that best capture the orthogonal drift vector are retrieved. In the second stage, these vocabulary items are used in a concept removal operation, producing embeddings that better align with the user's text prompt and serve as input for the final image generation.

**Step 1: Initial Image Generation.** First, an initial image $I$ is generated using the standard process. The text prompt $p_{\text{txt}}$ is encoded by two complementary text encoders: one produces a sequence of per-token embeddings $\mathbf{E}_p$, while the other yields a pooled sentence-level representation $\mathbf{e}_p$. Both conditioning signals are then provided to the image generator $\mathcal{G}$ to synthesize the initial image.

$$\mathbf{E}_p = \mathcal{E}_{\text{tok}}(p_{\text{txt}}), \qquad \mathbf{e}_p = \mathcal{E}_{\text{pool}}(p_{\text{txt}}), \qquad I = \mathcal{G}(\mathbf{E}_p, \mathbf{e}_p). \tag{3}$$

**Step 2: Semantic Drift Vector Calculation.** Next, we quantify the semantic discrepancy between the generated image $I$ and the original prompt $p_{\text{txt}}$. We use $\mathcal{E}_{\text{VLM}}$ to project both onto a shared embedding space. The *semantic drift vector*, $\mathbf{v}_{\text{drift}}$, captures the discrepancy between these two embeddings. This vector essentially encodes the semantic error, i.e., the direction and magnitude of the deviation of the image's content from the prompt's intent as perceived by the VLM.

$$\mathbf{v}_{\text{text}} = \mathcal{E}_{\text{VLM}}(p_{\text{txt}}), \qquad \mathbf{v}_{\text{img}} = \mathcal{E}_{\text{VLM}}(I), \qquad \mathbf{v}_{\text{drift}} = \mathbf{v}_{\text{img}} - \mathbf{v}_{\text{text}}. \tag{4}$$

**Step 3: Orthogonal Drift Component Isolation and Vocabulary Retrieval.** For $\mathbf{v}_{\text{drift}}$, we hypothesize that the most detrimental error components are those that introduce concepts extraneous to the prompt, which correspond to the direction orthogonal to the prompt's own embedding, $\mathbf{v}_{\text{text}}$. To isolate those components, we estimate the vector rejection of $\mathbf{v}_{\text{drift}}$ from $\mathbf{v}_{\text{text}}$. This is achieved by subtracting the projection of $\mathbf{v}_{\text{drift}}$ onto $\mathbf{v}_{\text{text}}$ from $\mathbf{v}_{\text{drift}}$ itself. The projection, $\mathbf{v}_{\text{proj}}$, is calculated as

$$\mathbf{v}_{\text{proj}} = \left( \frac{\mathbf{v}_{\text{drift}} \cdot \mathbf{v}_{\text{text}}}{\|\mathbf{v}_{\text{text}}\|^2} \right) \mathbf{v}_{\text{text}}. \tag{5}$$

The *orthogonal drift vector*, $\mathbf{v}_{\text{orth}}$, is then the rejection:

$$\mathbf{v}_{\text{orth}} = \mathbf{v}_{\text{drift}} - \mathbf{v}_{\text{proj}}. \tag{6}$$

To obtain a discrete, textual surrogate for $\mathbf{v}_{\text{orth}}$, we retrieve candidate terms from a custom vocabulary. To this end, we construct a large-scale vocabulary retrieval dataset designed to cover both visually grounded concepts and general linguistic phrases. The dataset construction process involves several stages of aggregation and filtering. We provide details of the dataset and its construction in Section 4.1. Each vocabulary item $w$ is associated with an embedding $\mathbf{e}_w$, and cosine similarities are computed as:

$$s(w) = \frac{\mathbf{v}_{\text{orth}} \cdot \mathbf{e}_w}{\|\mathbf{v}_{\text{orth}}\| \, \|\mathbf{e}_w\|}. \tag{7}$$

The top-$k$ terms with the highest similarity scores are selected, forming a discrete approximation of $\mathbf{v}_{\text{orth}}$. We denote this retrieved set of tokens as $\mathcal{S}_{\text{orth}}$.

**Step 4: Approximating Orthogonal Drift in Text Encoder Space.** The retrieved items $\mathcal{S}_{\text{orth}}$ are then re-embedded via the text encoders of the text-to-image model, yielding a per-token embedding matrix $\mathbf{E}_i$ and a pooled embedding vector $\mathbf{e}_i$ for each item $w_i \in \mathcal{S}_{\text{orth}}$. Since $\mathcal{S}_{\text{orth}}$ typically contains more than one item, we combine them using a softmax-weighted pooling to obtain equivalent representative embeddings:

$$\mathbf{E}_{\text{orth}} = \sum_i \beta_i \, \mathbf{E}_i, \qquad \mathbf{e}_{\text{orth}} = \sum_i \beta_i \, \mathbf{e}_i, \tag{8}$$

where the weights $\beta_i$ are defined by the normalized cosine similarity scores:

$$\beta_i = \frac{\exp(s(w_i))}{\sum_j \exp(s(w_j))}. \tag{9}$$

This results in a pair $(\mathbf{E}_{\text{orth}}, \mathbf{e}_{\text{orth}})$ that represents the surrogate set in the embedding space of the text encoders of our T2I model. These representative embeddings, in addition to the original embeddings, are supplied to the concept removal module.

**Step 5: Adaptive Rank-Reduced Concept Removal.** To remove undesired semantic concepts from the text embeddings, we employ an adaptive rank-reduction approach that automatically identifies and retains the meaningful semantic components while filtering out noise. We denote this operator by $\mathcal{R}$, and the procedure of its computation is summarized in Algorithm 2.

Given embedding tensors $\mathbf{E}_1, \mathbf{E}_2 \in \mathbb{R}^{B \times L \times H}$ (batch size $B$, sequence length $L$, hidden dimension $H$), we first determine the semantic rank of $\mathbf{E}_2$ through QR decomposition. For each batch element:

$$\mathbf{E}_2^T = \mathbf{Q}\mathbf{R}, \tag{10}$$

where $\mathbf{Q}$ contains orthonormal basis vectors. We identify semantically meaningful components by selecting basis vectors whose relative magnitude exceeds $\zeta = 1\%$ of the primary component:

$$k = \max\left\{ i : \frac{|r_{ii}|}{|r_{11}|} > \zeta \right\}, \tag{11}$$

where $r_{ii}$ are the diagonal elements of $\mathbf{R}$. We selected this threshold empirically to separate information from noise. Our analysis shows that while the majority of energy is concentrated in the first component, semantic distinctions are spread across several smaller components (ranks 3–15).

We then reconstruct a denoised version using only the top-$k$ basis vectors:

$$\mathbf{E}_2^{\text{reduced}} = (\mathbf{E}_2 \mathbf{Q}_{:,:k}) \mathbf{Q}_{:,:k}^T. \tag{12}$$

The concept removal operates on flattened representations to capture global semantic relationships. Let $\tilde{\mathbf{E}}_1 = \text{flatten}(\mathbf{E}_1)$ and $\tilde{\mathbf{E}}_2 = \text{flatten}(\mathbf{E}_2^{\text{reduced}})$. We compute:

$$\tilde{\mathbf{E}}_{\text{out}} = \tilde{\mathbf{E}}_1 - \alpha \, \text{Rej}_{\tilde{\mathbf{E}}_1}(\tilde{\mathbf{E}}_2) = \tilde{\mathbf{E}}_1 \cdot \left( 1 + \alpha \cdot \frac{\langle \tilde{\mathbf{E}}_1, \tilde{\mathbf{E}}_2 \rangle}{\|\tilde{\mathbf{E}}_1\|_2^2} \right) - \alpha \cdot \tilde{\mathbf{E}}_2, \tag{13}$$

where $\alpha$ controls the removal strength. The output is reshaped to the original dimensions. This adaptive approach automatically adjusts to concept complexity, i.e., simple concepts require fewer ranks while complex concepts utilize more, achieving robust removal without over-filtering.

**Algorithm 1** Orthogonal Drift Correction (ODC)

1: **Input:** Prompt $p_{\text{txt}}$, vocabulary $\mathcal{V}$
2: **Output:** Final image $I^*$
3: $(\mathbf{E}_p, \mathbf{e}_p) \leftarrow (\mathcal{E}_{\text{tok}}(p_{\text{txt}}), \mathcal{E}_{\text{pool}}(p_{\text{txt}}))$
4: $I \leftarrow \mathcal{G}(\mathbf{E}_p, \mathbf{e}_p)$
5: $\mathbf{v}_{\text{txt}} \leftarrow \mathcal{E}_{\text{VLM}}(p_{\text{txt}})$
6: $\mathbf{v}_{\text{img}} \leftarrow \mathcal{E}_{\text{VLM}}(I)$
7: $\mathbf{v}_{\text{drift}} \leftarrow \mathbf{v}_{\text{img}} - \mathbf{v}_{\text{txt}}$
8: $\mathbf{v}_{\text{proj}} \leftarrow \frac{\mathbf{v}_{\text{drift}} \cdot \mathbf{v}_{\text{txt}}}{\|\mathbf{v}_{\text{txt}}\|^2} \mathbf{v}_{\text{txt}}$
9: $\mathbf{v}_{\text{orth}} \leftarrow \mathbf{v}_{\text{drift}} - \mathbf{v}_{\text{proj}}$
10: **for** each $w \in \mathcal{V}$ **do**
11:     $\mathbf{e}_w \leftarrow \text{embedding}(w)$
12:     $s(w) \leftarrow \frac{\mathbf{v}_{\text{orth}} \cdot \mathbf{e}_w}{\|\mathbf{v}_{\text{orth}}\|\|\mathbf{e}_w\|}$
13: **end for**
14: $\mathcal{S}_{\text{orth}} \leftarrow$ Top-$k$ items by $s(w)$
15: Re-embed $\mathcal{S}_{\text{orth}}$: $\{(\mathbf{E}_i, \mathbf{e}_i)\}_{w_i \in \mathcal{S}_{\text{orth}}}$
16: Compute weights $\alpha_i \leftarrow \frac{\exp(s(w_i))}{\sum_j \exp(s(w_j))}$
17: $\mathbf{E}_{\text{orth}} \leftarrow \sum_i \alpha_i \mathbf{E}_i, \quad \mathbf{e}_{\text{orth}} \leftarrow \sum_i \alpha_i \mathbf{e}_i$
18: $(\mathbf{E}_{\text{refined}}, \mathbf{e}_{\text{refined}}) \leftarrow \mathcal{R}((\mathbf{E}_p, \mathbf{e}_p), (\mathbf{E}_{\text{orth}}, \mathbf{e}_{\text{orth}}))$
19: $I^* \leftarrow \mathcal{G}(\mathbf{E}_{\text{refined}}, \mathbf{e}_{\text{refined}})$
20: **return** $I^*$

**Algorithm 2** Adaptive Rank-Reduced Concept Removal ($\mathcal{R}$)

**Require:** $\mathbf{E}_1, \mathbf{E}_2 \in \mathbb{R}^{B \times L \times H}$, $\alpha \in [0,1]$, $\zeta = 0.01$; bounds $k_{\min} = 3$, $k_{\max} = 15$
**Ensure:** $\mathbf{E}_{\text{out}} \in \mathbb{R}^{B \times L \times H}$
1: **for** $b = 1$ to $B$ **do**
2:     $(\mathbf{Q}^{(b)}, \mathbf{R}^{(b)}) \leftarrow \text{QR\_decomposition}(\mathbf{E}_2^{(b)\top})$
3:     $\mathbf{r} \leftarrow \text{diag}(|\mathbf{R}^{(b)}|)$
4:     $k \leftarrow \max\{i : r_i/r_1 > \zeta\}$
5:     $k \leftarrow \text{clip}(k, k_{\min}, k_{\max})$
6:     Let $\mathbf{Q}_k^{(b)}$ be the first $k$ columns of $\mathbf{Q}^{(b)}$
7:     $\mathbf{E}_{2,\text{reduced}}^{(b)} \leftarrow (\mathbf{E}_2^{(b)} \mathbf{Q}_k^{(b)}) \mathbf{Q}_k^{(b)\top}$
8:     $\tilde{\mathbf{E}}_1 \leftarrow \text{flatten}(\mathbf{E}_1^{(b)})$
9:     $\tilde{\mathbf{E}}_2 \leftarrow \text{flatten}(\mathbf{E}_{2,\text{reduced}}^{(b)})$
10:     $\text{norm}_1 \leftarrow \|\tilde{\mathbf{E}}_1\|_2$
11:     $s \leftarrow 1 + \alpha \cdot \langle \tilde{\mathbf{E}}_1/\text{norm}_1, \tilde{\mathbf{E}}_2/\text{norm}_1 \rangle$
12:     $\tilde{\mathbf{E}}_{\text{out}} \leftarrow s \cdot \tilde{\mathbf{E}}_1 - \alpha \cdot \tilde{\mathbf{E}}_2$
13:     $\mathbf{E}_{\text{out}}^{(b)} \leftarrow \text{reshape}(\tilde{\mathbf{E}}_{\text{out}}, [L, H])$
14: **end for**
15: **return** $\mathbf{E}_{\text{out}}$

**Step 6: Final Image Generation.** Finally, a new image $I^*$ is generated using the refined conditioning embeddings $(\mathbf{E}_{\text{refined}}, \mathbf{e}_{\text{refined}})$ produced by the concept removal module. The image generator $\mathcal{G}$ then synthesizes:

$$I^* = \mathcal{G}(\mathbf{E}_{\text{refined}}, \mathbf{e}_{\text{refined}}). \tag{14}$$

By incorporating the correction derived from the surrogate embeddings, the refined conditioning better suppresses drift components while preserving the original semantic intent of the prompt.

## 4 Vocabulary Retrieval Dataset and COALBench-300

### 4.1 Construction of the Vocabulary Retrieval Dataset

To represent the isolated orthogonal drift vector $\mathbf{v}_{\text{orth}}$ using discrete textual surrogates, we constructed a vocabulary retrieval dataset of over 300,000 words and phrases designed to cover a diverse set of visually grounded and general linguistic concepts. The data was compiled from the following sources:

- **Open Images** (Krasin et al., 2017): Object class labels provided in the official release.

- **Visual Genome** (Krishna et al., 2017): Object annotations, which supply a wide range of free-form object names contributed by annotators.

- **Wikipedia** (Wikimedia Foundation, 2022): To augment the visual vocabulary with more general linguistic coverage, we parsed 1,000 articles extracted from English Wikipedia dataset (March 1, 2022 dump) using SpaCy's dependency parser (Honnibal et al., 2020). From each article, we extracted noun chunks, retaining those appearing at least three times across the corpus. This yielded a complementary set of multi-word expressions such as "machine learning" or "climate change."

In addition, we used a curated English word list (i.e., the `web2` dictionary via the `english-words` python package) to ensure broad lexical coverage (Wiens, 2025).

Following data collection from these sources, we apply a comprehensive preprocessing phase to clean and refine the dataset. This procedure consists of three primary stages: normalization, heuristic filtering, and spelling correction.

**Normalization.** All candidate strings were lowercased, Unicode-normalized, stripped of non-alphanumeric symbols, and reduced to tokens separated by single spaces. This step removed morphological noise such as capitalization, punctuation, and diacritics.

**Heuristic filtering.** To eliminate uninformative entries, we applied a series of heuristic rules. First, we enforced length constraints, retaining only terms that contain 1 to 3 tokens and span between 3 and 30 characters. Next, we discarded strings dominated by digits or punctuation, as well as isolated stopword phrases (e.g., "and", "the"). Finally, to ensure that short but meaningful concepts were not mistakenly removed by these length constraints, we implemented a whitelist to preserve essential terms such as "tv" and "pc".

**Spelling correction.** Because crowdsourced annotations introduce spelling inconsistencies, we employed the SymSpell algorithm (Garbe, 2025) with a maximum edit distance of 2 to canonicalize misspellings. This reduced duplication from typographic errors.

This multi-stage process produced a broad and diverse vocabulary containing both concrete, visually grounded object names and higher-level conceptual phrases. In total, the final vocabulary contains approximately 323,000 items, which serves as the vocabulary retrieval dataset for our experiments.

## 4.2 Construction of CoALBench-300

In order to evaluate the semantic alignment between text-to-image models and user intent, we created CoALBench-300 (**CO**mpositional **AL**ignment Benchmark), a curated collection of 300 imaginative prompts specifically designed to test challenging semantic alignment scenarios not emphasized in existing benchmarks. The goal was to establish a standardized set of inputs characterized by richness and diversity such that alignment failures would be clearly observable. To this end, we queried three distinct large language models: OpenAI o3 (`o3-2025-04-16`), Anthropic Claude Opus-4 (`claude-opus-4-20250514`), and Google Gemini 2.5 Pro (`gemini-2.5-pro`), with instructions to propose creative and highly visual text-to-image prompts. Each model produced a large pool of candidate prompts emphasizing diverse styles, scenarios, and semantic complexity.

From the generated pool, we manually reviewed and curated a final set of 300 prompts. This final collection of curated prompts provides a compact but challenging benchmark that allows us to systematically probe how well text-to-image models respect compositional semantics. Unlike generic caption corpora, these prompts emphasize stress-testing of alignment through imaginative and specific requests. The resulting dataset is presented in Table A1 of Appendix A.

## 5 Experiments

### 5.1 Experimental Setup

**Models.** To demonstrate the model-agnostic nature of ODC, we test it across Stable Diffusion XL (SDXL) (Podell et al., 2023) and FLUX.1 [schnell] (Black Forest Labs, 2024). We use BLIP2-ITM-ViT-G model (Li et al., 2023) for vocabulary retrieval in the first stage of ODC.

**Inference Parameters.** In our experiments, we generated images at a resolution of 1024×1024 pixels. For SDXL, we used the Euler Discrete sampler with 30 inference steps and a Classifier-Free Guidance (CFG) scale of 7.5. For FLUX.1 [schnell], we employed the Flow Matching Euler Discrete sampler with 4 inference steps. To ensure statistical robustness, we generated three images per prompt using fixed seeds across all methods and report the mean values for all metrics. Our method introduces a few hyperparameters, which are empirically set for inference-time guidance, rather than relying on tuning the model. Specifically, we

Table 1: Main quantitative results on CoalBench-300 and PartiPrompts benchmarks. We report alignment metrics across Stable Diffusion XL and FLUX.1 [schnell]. Best results are in **bold**. Improvements are shown in **green percentages** relative to the Vanilla baseline. Blue and Orange rows highlight our methods. The best-of-two variant (ODC-BoT) selects the best image from the two available choices—see text for details.

| Method | CoalBench-300 | | | | PartiPrompts | | | |
|---|---|---|---|---|---|---|---|---|
| | Stable Diffusion XL | | FLUX.1 [schnell] | | Stable Diffusion XL | | FLUX.1 [schnell] | |
| | CLIP ↑ | BLIP ↑ | CLIP ↑ | BLIP ↑ | CLIP ↑ | BLIP ↑ | CLIP ↑ | BLIP ↑ |
| Vanilla | 30.305 | 0.608 | 29.262 | 0.608 | 30.834 | 0.603 | 31.239 | 0.605 |
| CFG-Sweep (CFG=5.0) | 29.953 | 0.607 | - | - | 30.447 | 0.602 | - | - |
| CFG-Sweep (CFG=12.0) | 30.392 | 0.608 | - | - | 31.127 | 0.604 | - | - |
| Negative Prompt | 30.305 | 0.608 | 29.296 | 0.608 | 30.834 | 0.603 | 31.184 | 0.605 |
| **ODC (Ours)** | 31.741 +4.74% | 0.612 +0.66% | 29.612 +1.20% | 0.608 | 32.503 +5.41% | 0.607 +0.66% | 31.451 +0.68% | 0.605 |
| **ODC-BoT (Ours)** | **32.205 +6.27%** | **0.613 +0.82%** | **30.4 +3.89%** | **0.61 +0.33%** | **32.99 +6.99%** | **0.608 +0.83%** | **32.268 +3.29%** | **0.607 +0.33%** |

retrieved the top $k = 5$ vocabulary items, applied softmax pooling with temperature 0.5, and set the concept removal weighting parameter $\alpha$ to 1.0 for SDXL and 0.5 for FLUX.1 [schnell].

**Benchmarks and Datasets.** We evaluate our method using three distinct sets of prompts. First, we employ CoalBench-300, introduced in Section 4.2. Second, we utilize PartiPrompts (Yu et al., 2022) to probe imaginative and open-ended generation. For our experiments, we filter this dataset to retain only prompts containing ten words or more, ensuring that all evaluated prompts provide sufficient semantic grounding. Lastly, we employ T2I-CompBench (Complex) (Huang et al., 2025), a challenging benchmark subset that evaluates the generation of complex scenes involving multiple objects, attributes, and relationships. Experiments with these datasets present a thorough evaluation of the methods.

**Baselines.** We compare ODC against several baselines to better understand its performance. We use the unmodified output from the base T2I models as our Vanilla reference. We also perform a CFG-Sweep, reporting results across different scales to compare against simply strengthening prompt guidance. Additionally, we evaluate a Negative Prompt baseline, where we create a tailored negative prompt for each generation by joining retrieved vocabulary items, using it in place of generic negative prompts (e.g., 'bad anatomy, extra fingers, blurry'). Finally, we compare against two specific inference-time guidance approaches, Attend-and-Excite (Chefer et al., 2023), which refines the latent code during generation by optimizing cross-attention units to "excite" neglected subject tokens (adapted for SDXL; see Appendix B), and ToMe (Hu et al., 2024), which merges object and attribute embeddings into composite tokens to unify their cross-attention maps via iterative optimization and entropy regularization.

**Evaluation Metrics.** We use established metrics for text-image alignment and computational efficiency. To measure semantic alignment between a prompt and its generated image, we rely on both CLIPScore (ViT-L/14) (Hessel et al., 2021), a widely used metric for general semantic similarity, and BLIPScore (BLIP2-ITM-ViT-G) (Li et al., 2023), which leverages the more advanced BLIP-2 vision-language model for finer-grained evaluations. Additionally, for the T2I-CompBench (Complex) benchmark, we report the "3-in-1" score alongside the individual component scores (BLIP-VQA, UniDet, and CLIPScore) to assess performance across attribute binding, spatial arrangement, and semantic alignment. Given that our method operates during inference, we measure its computational overhead in terms of both latency, defined as the average wall-clock time required to generate a single image, and GPU memory usage during generation.

## 5.2 Quantitative Results

To establish the efficacy of our method, we present a thorough quantitative evaluation. As shown in Tables 1 and 2, ODC consistently and significantly outperforms all baselines across all faithfulness metrics on both Stable Diffusion XL and FLUX.1 [schnell].

Table 2: Comparison with state-of-the-art. We report CLIP and BLIP scores for CoALBench-300 and PartiPrompts benchmarks. For T2I-CompBench, we report the metrics on the complex subset, including the final 3-in-1 average score. Best results are in **bold**. All methods use SDXL as the baseline.

| Method | CoALBench-300 | | PartiPrompts | | T2I-CompBench (Complex) | | | |
|---|---|---|---|---|---|---|---|---|
| | CLIP ↑ | BLIP ↑ | CLIP ↑ | BLIP ↑ | BLIP-VQA ↑ | UniDet ↑ | CLIP ↑ | 3-in-1 ↑ |
| SDXL (Podell et al., 2023) | 30.305 | 0.608 | 30.834 | 0.603 | 0.442 | **0.025** | 0.293 | 0.3294 |
| A&E (Chefer et al., 2023) | 29.542 | 0.608 | 30.15 | 0.604 | 0.436 | 0.023 | 0.290 | 0.3259 |
| ToMe (Hu et al., 2024) | 30.397 | 0.608 | 30.681 | 0.603 | 0.468 | 0.022 | 0.292 | 0.3417 |
| **ODC (Ours)** | **31.741** +4.74% | **0.612** +0.66% | **32.503** +5.41% | **0.607** +0.66% | **0.494** +11.76% | 0.020 | **0.300** +2.39% | **0.3572** +8.44% |

Across all evaluations and for both model backbones (Stable Diffusion XL and FLUX.1 [schnell]), ODC consistently improves semantic alignment scores compared to existing baselines. Since ODC naturally produces both the original baseline image and the corrected variant, we can adopt a simple best-of-two (ODC-BoT) strategy by returning whichever image achieves the higher score. This strategy yields improved results with negligible computational overhead.

Table 3: Efficiency analysis on SDXL.

| Method | Latency (s/img) | Memory (GB) |
|---|---|---|
| Vanilla | 3.637 | 23.015 |
| ODC | 7.560 | 23.015 |

As shown in Table 3, ODC introduces some latency overhead while leaving memory consumption unchanged. Compared to the vanilla baseline, latency increases from 3.637 s/img to 7.56 s/img, reflecting the additional correction pass. Importantly, GPU memory usage remains constant at 23.015 GB, making the approach just as accessible in practice as the baseline methods. Overall, these results demonstrate that the gains in semantic alignment come with predictable and manageable efficiency costs.

### 5.3 Qualitative Results

Visual inspection of the generated images provides the most intuitive evidence of ODC's effectiveness. In Figure 2, we present a side-by-side comparison of images generated by vanilla SDXL and FLUX and our ODC-corrected method for a variety of challenging prompts.

We observe that ODC improves compositional accuracy and attribute binding, and leads to reduced semantic drift. The baseline often struggles with correct modifier-object relationships and maintaining fine-grained attributes, while ODC generates images that align more accurately with the user description. Furthermore, for prompts that are vulnerable to spurious additions (e.g., "Victorian astronaut playing violin inside ornate greenhouse orbiting Saturn's rings"), the baseline model introduces unintended elements, such as additional violinists, that reduce alignment with the user prompt. In contrast, ODC constrains generations to the requested content and avoids unwanted additions, yielding outputs that more faithfully reflect the intended semantics. These qualitative examples demonstrate that by refining the initial text embedding, ODC provides a more reliable conditioning for the image generation process. Additional qualitative results are provided in Appendix C.

### 5.4 Ablation Studies

**The Importance of Orthogonal Correction.** Our central claim is that isolating and removing the *orthogonal* component of the semantic drift is key. To test this, we implemented a variant of our method, "Full Vector Correction," which uses the whole semantic drift vector ($\mathbf{v}_{\mathrm{drift}}$) to retrieve the top-k vocabulary items. As shown in Table 4 and Figure 3, while Full Vector Correction provides an improvement over the baseline, it underperforms our proposed ODC. This result strongly

Table 4: Quantitative ablation results for SDXL on CoALBench-300.

| Method | CLIP ↑ |
|---|---|
| Vanilla | 30.305 |
| Full Vector Correction | 31.489 |
| Orthogonal Drift Correction | **31.741** |

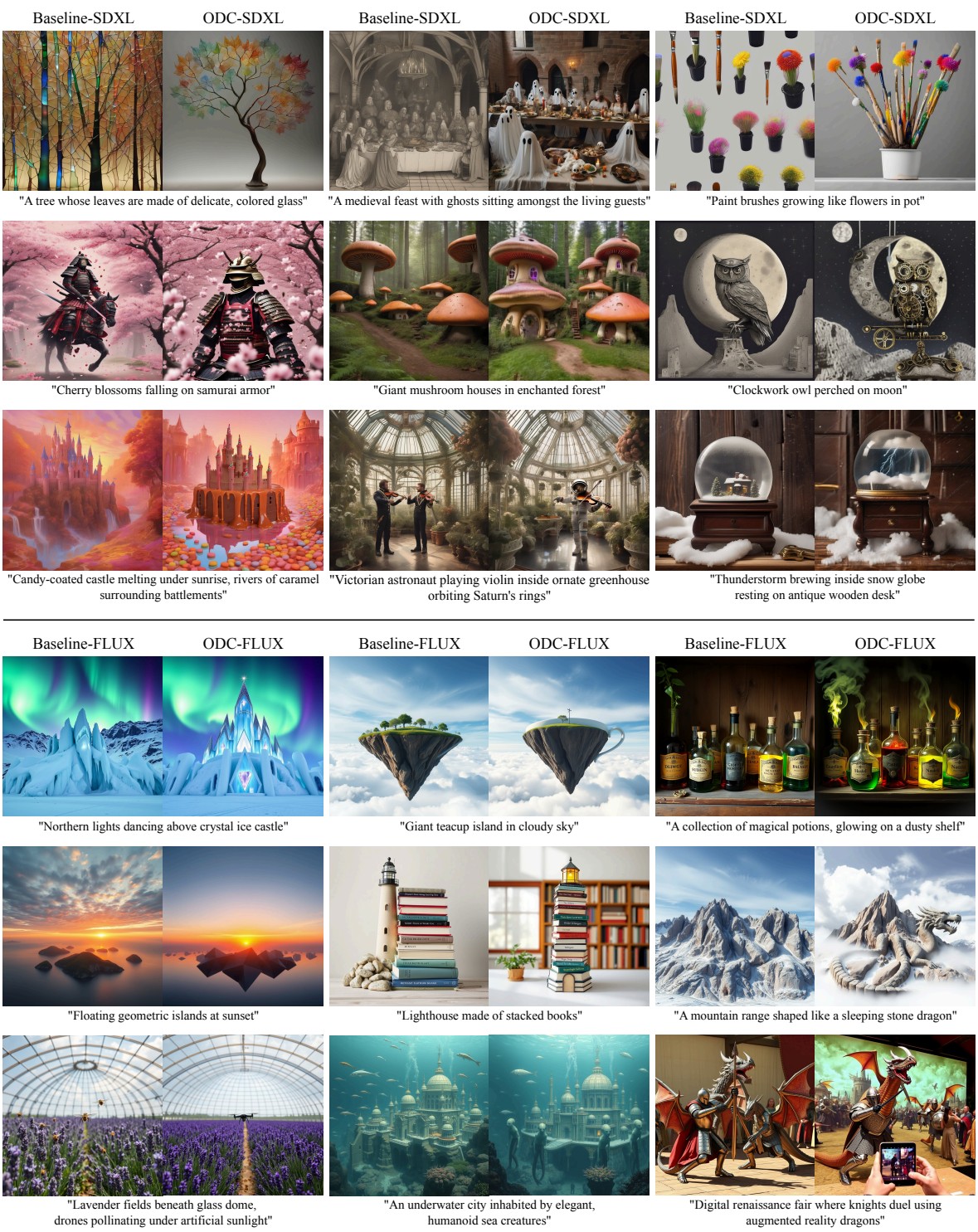

Figure 2: Qualitative comparison of images generated with and without Orthogonal Drift Correction (ODC). For each prompt, we show the output from the baseline model (left) and our ODC-corrected version (right).

supports our hypothesis. Subtracting the full vector likely over-corrects by removing useful on-axis semantic information, whereas ODC surgically removes only the irrelevant semantic contents.

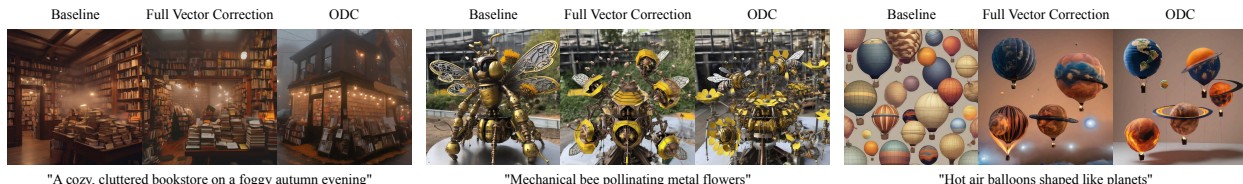

| Baseline | Full Vector Correction | ODC | Baseline | Full Vector Correction | ODC | Baseline | Full Vector Correction | ODC |

"A cozy, cluttered bookstore on a foggy autumn evening"    "Mechanical bee pollinating metal flowers"    "Hot air balloons shaped like planets"

Figure 3: Qualitative ablation study results showing three example prompts. For each example, we compare the base model (SDXL), Full Vector Correction, and proposed ODC. The results indicate that ODC corrects the semantic discrepancies observed in the left and middle columns.

## 6  Limitations

Despite its strong performance, ODC has some inherent limitations. First, its effectiveness is bound by the perceptual capabilities of the vision-language model (VLM) used in the first stage. If the VLM cannot see a specific semantic error, ODC may not correct it. This is most apparent with highly complex and abstract concepts, which remain challenging for current VLM embeddings to represent accurately. However, with the rapid advances in VLMs research, this limitation is expected to gradually fade away.

Second, the two-stage correction process inherently increases the inference latency compared to a standard single pass, as shown in Table 3. While the overhead from the embedding calculations themselves is negligible, the need for an initial generation pass represents a direct trade-off between computational cost and semantic drift correction. Nevertheless, considering that T2I generation is generally an offline task, this limitation is of minor nature. Moreover, our method can potentially reduce the time associated with manual trial and error process involved in prompt engineering considerably by providing better alignment of outputs to the prompts. Having this ability without requiring additional memory makes our method pragmatically appealing.

## 7  Conclusion

In this paper, we addressed the critical problem of semantic drift in text-to-image models. We introduced Orthogonal Drift Correction (ODC), a training-free, inference-time method that guides the image generation through a two-stage process. It estimates the components of text embeddings causing the drift and mitigates them. Our extensive experiments demonstrate that ODC significantly enhances prompt-image alignment across multiple models and benchmarks. Without requiring any retraining or architectural changes, ODC provides a practical and generalizable solution for improving the alignment and reliability of existing pre-trained text-to-image models, making them more powerful creative tools.

### Reproducibility Statement

To ensure reproducibility, we provide detailed algorithmic descriptions of our method in Section 3, along with hyperparameter configurations and experimental settings in Section 5. Our proposed prompt dataset, CoALBench-300, is provided in full in Appendix A, and all other datasets used in our experiments are publicly available. All experiments are conducted using publicly available models and libraries. We intend to release our code publicly in the future.

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

## A    COALBench-300

Table A1 presents the CoALBench-300 dataset used in our experiments, consisting of 300 prompts generated as described in Section 4.2.

Table A1: Full list of prompts in CoALBench-300.

| # | Prompt | # | Prompt |
|---|--------|---|--------|
| 1 | Neon cityscape reflected in rain puddles at midnight | 2 | Ancient library filled with floating glowing books |
| 3 | Steampunk octopus playing violin underwater | 4 | Cherry blossoms falling on samurai armor |
| 5 | Cyberpunk cat cafe with holographic menu displays | 6 | Dragon made entirely of autumn leaves |
| 7 | Astronaut planting flowers on Mars surface | 8 | Victorian greenhouse overflowing with bioluminescent plants |
| 9 | Robot chef preparing sushi in minimalist kitchen | 10 | Northern lights dancing above crystal ice castle |
| 11 | Abandoned amusement park reclaimed by nature | 12 | Phoenix rising from coffee cup steam |
| 13 | Underwater city inside giant glass bubble | 14 | Mechanical butterfly landing on child's outstretched hand |
| 15 | Storm clouds shaped like galloping horses | 16 | Art deco train station at golden hour |
| 17 | Jellyfish constellation floating through space | 18 | Medieval knight riding mechanical dragon |
| 19 | Sunflowers growing from piano keys | 20 | Time traveler's workshop filled with clocks |
| 21 | Magical forest where trees have doorways | 22 | Origami birds transforming into real ones |
| 23 | Vintage typewriter typing rainbow colored words | 24 | Giant teacup island in cloudy sky |
| 25 | Bioluminescent mushroom cave with crystal formations | 26 | Retro diner floating in outer space |
| 27 | Library ladder reaching into starry sky | 28 | Glass sculpture garden during thunderstorm |
| 29 | Mechanical whale swimming through clouds | 30 | Paint brushes growing like flowers in pot |

Table A1 – *Continued from previous page*

| # | Prompt | # | Prompt |
|---|--------|---|--------|
| 31 | Lighthouse made of stacked books | 32 | Geometric mountains with gradient sunset colors |
| 33 | Tiny fairy village inside terrarium | 34 | Steam locomotive made of flowers |
| 35 | Crystal skull containing miniature galaxy | 36 | Hot air balloons shaped like planets |
| 37 | Underwater ballet performed by seahorses | 38 | Gothic cathedral made of ice |
| 39 | Neon sign reflected in cyberpunk alley | 40 | Tree house connected by rope bridges |
| 41 | Desert oasis with mirror-like water | 42 | Mechanical heart with gears and flowers |
| 43 | Floating islands connected by waterfalls | 44 | Vintage camera capturing memories as butterflies |
| 45 | Snow globe containing miniature city | 46 | Giant mushroom houses in enchanted forest |
| 47 | Clock tower melting like Salvador Dali painting | 48 | Paper airplane leaving trail of stars |
| 49 | Underwater ruins with glowing hieroglyphs | 50 | Mechanical bee pollinating metal flowers |
| 51 | Aurora borealis inside crystal cave | 52 | Sailing ship made of autumn leaves |
| 53 | Robot gardener tending glass flowers | 54 | Mountain peak piercing through clouds |
| 55 | Neon Tokyo street in heavy rain | 56 | Butterfly wings made of stained glass |
| 57 | Ancient tree with doors to other worlds | 58 | Mechanical dragon breathing rainbow fire |
| 59 | Floating market on lily pad boats | 60 | Crystal palace reflecting sunset colors |
| 61 | Origami crane transforming into phoenix | 62 | Underwater volcano with lava lamp effect |
| 63 | Clockwork owl perched on moon | 64 | Garden maze viewed from above |
| 65 | Holographic whale swimming through city | 66 | Ice sculpture melting into butterflies |
| 67 | Vintage bicycle with flower basket | 68 | Portal opening in library wall |
| 69 | Mechanical spider weaving light web | 70 | Treehouse city in giant redwood forest |
| 71 | Glass elevator ascending through clouds | 72 | Firefly jar illuminating dark forest |
| 73 | Geometric deer made of neon lines | 74 | Floating tea party in sky |
| 75 | Crystal bridge over mirror lake | 76 | Mechanical hummingbird feeding on gear flowers |
| 77 | Northern lights reflected in frozen lake | 78 | Paper boat sailing on book pages |
| 79 | Bonsai tree with tiny glowing lanterns | 80 | Underwater castle made of coral |

*Continued on next page*

Table A1 – *Continued from previous page*

| # | Prompt | # | Prompt |
|---|--------|---|--------|
| 81 | Steampunk hot air balloon city | 82 | Galaxy swirling in coffee cup |
| 83 | Ice phoenix rising from frozen waterfall | 84 | Mechanical butterfly collection in display case |
| 85 | Floating geometric islands at sunset | 86 | Glass greenhouse in snowy landscape |
| 87 | Lightning striking sand creating glass | 88 | Origami dragon breathing paper fire |
| 89 | Crystal cave with rainbow reflections | 90 | Vintage train crossing bridge at dawn |
| 91 | Mechanical whale breaching ocean surface | 92 | Fairy ring mushrooms glowing at twilight |
| 93 | Neon arcade in rain-soaked alley | 94 | Tree growing through abandoned car |
| 95 | Holographic fish swimming through air | 96 | Ice palace under aurora borealis |
| 97 | Steampunk telescope viewing distant galaxies | 98 | Garden path made of piano keys |
| 99 | Crystal butterfly landing on metal flower | 100 | Lighthouse beam cutting through fog |
| 101 | A bioluminescent jellyfish floating through a neon-lit cyberpunk city | 102 | A lone knight overlooking a valley of giant, glowing mushrooms |
| 103 | A crystal skull filled with a swirling, miniature galaxy | 104 | A serene, minimalist Japanese garden in the pouring rain |
| 105 | An ancient Roman library on a terraformed, dusty Mars | 106 | A tiny fairy house built inside a broken teapot |
| 107 | A grand piano with keys melting like liquid Dali-style | 108 | A cozy, cluttered bookstore on a foggy autumn evening |
| 109 | A Viking longship sailing on a sea of clouds | 110 | A stained glass window depicting a cosmic horror entity |
| 111 | A sleek, organic starship gliding silently past a vibrant nebula | 112 | A grizzly bear made of constellations in the night sky |
| 113 | A forgotten, overgrown greenhouse filled with strange, glowing plants | 114 | A city skyline at sunset, rendered in 8-bit pixel art |
| 115 | A steampunk inventor's workshop, cluttered with brass and clockwork | 116 | A macro photograph of a single dewdrop on a spiderweb |
| 117 | A majestic castle made of obsidian and jagged lightning | 118 | The concept of "silence" visualized as a physical object |
| 119 | An Art Deco robot butler serving cocktails at a party | 120 | A floating market on a serene, otherworldly alien canal |
| 121 | A portrait of a queen, made entirely from pressed flowers | 122 | A paper-craft, origami world with a flowing paper river |
| 123 | A cat made of glass, filled with swirling, colored smoke | 124 | A haunted forest where the trees have screaming human faces |
| 125 | Impossible Escher-like architecture carved from a single, giant tree | 126 | A gourmet hamburger made entirely of sparkling, edible gemstones |

Table A1 – *Continued from previous page*

| # | Prompt | # | Prompt |
|---|--------|---|--------|
| 127 | A desert landscape with three suns setting on the horizon | 128 | An ancient, moss-covered golem sleeping in a sunlit forest |
| 129 | A city street scene painted in the style of Van Gogh | 130 | A lone, red telephone box in the middle of nowhere |
| 131 | A mechanical heart with glowing tubes and intricate gears | 132 | A tranquil koi pond that reflects a starry night sky |
| 133 | An astronaut discovering an ancient alien ruin on the moon | 134 | A single, perfect rose encased in a block of ice |
| 135 | An island that is the back of a giant, swimming turtle | 136 | A shadowy figure standing at the end of a long pier |
| 137 | A black and white photograph of a 1940s detective's office | 138 | A city built inside a colossal, hollowed-out crystal cavern |
| 139 | A samurai warrior with glowing, cybernetic neon tattoos | 140 | A massive, ancient library where the books float between shelves |
| 141 | A cup of coffee with a swirling galaxy in it | 142 | A child releasing a kite that is actually a dragon |
| 143 | A detailed anatomical drawing of a mythical griffin | 144 | A vaporwave sculpture garden with pink and blue neon lights |
| 145 | A doorway to another dimension opening in a brick wall | 146 | A city street dissolving into musical notes in the air |
| 147 | A lone lighthouse battered by a furious, cosmic storm | 148 | A portrait of "joy" as an explosion of vibrant colors |
| 149 | A medieval feast with ghosts sitting amongst the living guests | 150 | An underwater city inhabited by elegant, humanoid sea creatures |
| 151 | A tree whose leaves are made of delicate, colored glass | 152 | A futuristic high-speed train blurring through a bamboo forest |
| 153 | A sad clown looking at his reflection in a puddle | 154 | A cinematic shot of a dinosaur roaring in a modern city |
| 155 | A map of a fantasy world burned onto old leather | 156 | An ornate, Venetian mask with intricate filigree and feathers |
| 157 | A car rusting in a field of blooming sunflowers | 158 | A mountain range shaped like a sleeping stone dragon |
| 159 | A wizard's spellbook lying open, glowing with powerful magic | 160 | A skeleton in a spacesuit, sitting on a desolate asteroid |
| 161 | A beautiful landscape painting done on a crumbling concrete wall | 162 | A ceramic mosaic depicting the birth of a star |
| 163 | A clock tower where the numbers are flying away | 164 | An elven city built seamlessly into giant, living trees |
| 165 | A low-poly render of a wolf howling at the moon | 166 | A secret garden hidden behind a waterfall |
| 167 | A single, burning feather falling from a dark sky | 168 | A long exposure photograph of highway lights at night |

*Continued on next page*

Table A1 – *Continued from previous page*

| # | Prompt | # | Prompt |
|---|--------|---|--------|
| 169 | A palace made entirely of ice, glistening in the sun | 170 | A chessboard with living pieces preparing for battle |
| 171 | A surreal landscape where the ground is a reflective mirror | 172 | A powerful phoenix rising from ashes, cinematic fantasy art |
| 173 | A ukiyo-e woodblock print of a busy Tokyo intersection | 174 | An old, forgotten god statue entangled in jungle vines |
| 175 | A gentle robot tending to a garden of metal flowers | 176 | An underwater submarine discovering the lost city of Atlantis |
| 177 | The solar system re-imagined as a delicate, hanging mobile | 178 | A tiny mouse wearing intricate, hand-crafted medieval armor |
| 179 | A world where the oceans are in the sky | 180 | A close-up of a dragon's eye, reflecting a battle |
| 181 | A hyper-realistic painting of a spilled glass of milk | 182 | A towering, complex sandcastle about to be hit by a wave |
| 183 | A lush, tropical beach inside a giant, open clam shell | 184 | A synthwave-style sunset over a retro-futuristic highway |
| 185 | A collection of magical potions, glowing on a dusty shelf | 186 | A double exposure portrait of a woman and a forest |
| 187 | A pirate ship sailing through a massive electrical storm | 188 | A whimsical town where all the buildings are books |
| 189 | A soldier's helmet from the future, cracked and abandoned | 190 | A tree of life connecting the earth and the cosmos |
| 191 | A blueprint-style technical drawing of a fantasy airship | 192 | A labyrinth made of tall, perfectly trimmed hedges |
| 193 | An abstract sculpture representing the sound of laughter | 194 | A street market on a space station, full of aliens |
| 195 | A beautiful, haunting portrait of a porcelain doll | 196 | The world balanced on the tip of a giant's finger |
| 197 | A polar bear sleeping under the vibrant aurora borealis | 198 | A vintage 1950s car flying through the clouds |
| 199 | A stone staircase spiraling down into the deep, dark ocean | 200 | A single, glowing key lying on an empty city street |
| 201 | Neon samurai meditating under cherry blossoms, cyberpunk city distant horizon | 202 | Golden dragon soaring above tempestuous sea during crimson twilight storm |
| 203 | Victorian astronaut playing violin inside ornate greenhouse orbiting Saturn's rings | 204 | Gigantic marble statues emerging from desert dunes beneath aurora sky |
| 205 | Steam-powered whale floating over steampunk city releasing glowing lanterns peacefully | 206 | Surreal library labyrinth with endless staircases and levitating luminous books |

Table A1 – *Continued from previous page*

| # | Prompt | # | Prompt |
|---|--------|---|--------|
| 207 | Winter fox spirits dancing around ancient shrine under full moon | 208 | Retro-futuristic diner on Mars serving milkshakes to robot cowboys today |
| 209 | Bioluminescent jungle river reflecting twin moons, explorers paddling crystal canoes | 210 | Minimalist ink painting depicting thunderstorm over solitary mountain tea house |
| 211 | Ancient turtle island carrying cherry forest drifting across tranquil ocean | 212 | Post-apocalyptic fashion runway among ruins, models wearing repurposed metallic couture |
| 213 | Glass butterflies emerging from cracked sculpture in abandoned classical gallery | 214 | Cosmic carnival Ferris wheel spinning galaxies instead of passenger gondolas |
| 215 | Medieval alchemist laboratory illuminated by swirling emerald and violet potions | 216 | Peaceful Zen garden floating among clouds, koi fish swimming air |
| 217 | Art deco submarine cruising crystal Arctic waters beneath shimmering aurora | 218 | Gothic cathedral built from bones, candles flickering with blue flames |
| 219 | Whimsical cat orchestra performing symphony inside giant teacup on meadow | 220 | Autumn maple leaves transforming into origami cranes mid gentle breeze |
| 221 | Futuristic Venice with glass canals and levitating gondolas at sunset | 222 | Alien botanist tending glowing succulents within transparent asteroid greenhouse sphere |
| 223 | Samurai mech duel atop skyscraper during torrential rain and lightning | 224 | Childhood memories visualized as floating polaroid cubes above grassy field |
| 225 | Mushroom village illuminated by fireflies, tiny gnomes celebrating midsummer feast | 226 | Surreal chessboard landscape where pieces are skyscrapers casting elongated shadows |
| 227 | Golden hour lighthouse standing on clouds guiding paper boats home | 228 | Cybernetic phoenix reborn from neon flames above dystopian alleyway rooftops |
| 229 | Mythical underwater library with coral shelves and manta ray librarians | 230 | Celtic wolf warrior howling beside rune-carved stone under starry sky |
| 231 | Industrial robots painting classical renaissance portrait inside abandoned factory hall | 232 | Pastel vaporwave beach scene with dolphins and pixelated palm trees |
| 233 | Ice cream planet orbiting candy stars within childlike cosmic dreamscape | 234 | Ethereal ballerina composed of smoke gracefully twirling through candlelit ballroom |
| 235 | Ancient astronauts carved into mountain, projections igniting during equinox ceremony | 236 | Monochrome noir city where rain turns neon once striking pavement |

*Continued on next page*

Table A1 – *Continued from previous page*

| # | Prompt | # | Prompt |
|---|--------|---|--------|
| 237 | Giant panda knight wearing bamboo armor charging across sunflower plains | 238 | Sailing ship overhead sky ocean, clouds shaped like wandering whales |
| 239 | Magnificent art nouveau dragonfly airship docking at crystal palace port | 240 | Crimson haired witch brewing constellation soup in antique cast-iron cauldron |
| 241 | Time travelers picnic within looping paradox meadow filled with suns | 242 | Dreamy pastel clouds dripping like paint over sleepy coastal village |
| 243 | Crystal caverns resonating music, stalactites functioning as grand pipe organ | 244 | Pharaoh cyborg sitting upon holographic throne inside starship pyramid command |
| 245 | Origami dragons battling above waterfall, paper scraps becoming swirling mist | 246 | Intergalactic farmers harvesting stardust using laser scythes and hover wagons |
| 247 | Snowy owl postman delivering letters between lonely mountain observatories nightly | 248 | Renaissance painting fused with glitch art, saints pixelated yet divine |
| 249 | Lonely astronaut gardening roses inside cracked helmet on barren exoplanet | 250 | Midsummer carnival mirrored on lake surface, reflections showing alternate dimension |
| 251 | Celestial koi swimming through starry void forming luminous cosmic patterns | 252 | Gigabyte forest where data streams cascade like waterfalls between trees |
| 253 | Clockwork hummingbirds pollinating gears within enormous steampunk garden mechanism today | 254 | Futuristic samurai riding holographic horse across neon rice paddies night |
| 255 | Survivalists camping on colossal lily pads floating down moonlit river | 256 | Hyperrealistic microphotography style city built upon a dandelion seed closeup |
| 257 | Norwegian fjord village illuminated solely by bioluminescent algae and stars | 258 | Astral projection commuters riding light beams between futuristic office towers |
| 259 | Fairytale bakery selling clouds, customers bottling rainbows for frosting today | 260 | Desert caravan of holographic camels crossing dunes under binary suns |
| 261 | Spaceship graveyard turned skatepark, teenagers grinding along rusted hulls today | 262 | Fantasy opera house shaped like seashell, audience wearing pearl masks |
| 263 | Interactive graffiti walls repaint themselves according to passerby's emotional aura | 264 | Robot monk studying lotus algorithm amidst tranquil chrome bamboo grove |
| 265 | Abandoned amusement park reclaimed by jungle, rollercoaster entwined with vines | 266 | Crystalline wolf pack running across frozen lake under kaleidoscope aurora |

*Continued on next page*

Table A1 – *Continued from previous page*

| # | Prompt | # | Prompt |
|---|--------|---|--------|
| 267 | Pocket watch melting into river, time flowing downstream with salmon | 268 | Digital renaissance fair where knights duel using augmented reality dragons |
| 269 | Samurai standing in field of wind turbines during golden sunrise | 270 | Galactic mail carrier surfing solar winds to deliver interstellar postcards |
| 271 | Carousel horses awakening midnight, galloping across city streets leaving sparks | 272 | Northern lights shaped like dragons over mirrorlike fjord with rowboat |
| 273 | Cyberpunk mermaid lounging on neon coral reef playing electric harp | 274 | Snow globe metropolis shaking, skyscraper flakes swirling around central park |
| 275 | Mystical doorway inside tree trunk leading to floating candlelit staircase | 276 | Marble chess pieces crying tears, puddles reflecting parallel universe conflicts |
| 277 | Steampunk octopus captain steering dirigible through thunderous cloud ocean today | 278 | Silk road marketplace set on asteroid, merchants trading gravity spices |
| 279 | Lone monk meditating atop newspaper stack skyscraper in bustling metropolis | 280 | Candy-coated castle melting under sunrise, rivers of caramel surrounding battlements |
| 281 | Moonlit bamboo forest where shadows of past lives perform theatre | 282 | Quantum computer envisioned as cathedral, qubits glowing like stained glass |
| 283 | Vibrant coral reef mirrored above water, sky filled with fish | 284 | Ghostly steam train crossing galaxy, passengers silhouettes made from stardust |
| 285 | Medieval tavern hosting extraterrestrial travelers swapping holographic treasure maps tonight | 286 | Arcade cabinet forest, joysticks sprouting like mushrooms among pixel leaves |
| 287 | Viking ship sailing across clouds towards colossal floating mead hall | 288 | Haunted typewriter printing poems by itself inside candlelit attic loft |
| 289 | Interdimensional bus stop where travelers trade memories for tickets home | 290 | Mid-century modern house perched atop giant redwood, overlooking fog valley |
| 291 | Lavender fields beneath glass dome, drones pollinating under artificial sunlight | 292 | Ancient rune compass projecting holographic maps above weathered pirate desk |
| 293 | Swan boat gliding through clouds, passengers sipping tea with angels | 294 | Neo-Tokyo skyline reflected in raindrop on cybernetic geisha's finger nail |
| 295 | Prismatic desert canyon walls refracting sunlight into rainbow patterned sand | 296 | Clocktower orchard where apples tick loudly with winding golden gears |
| 297 | Giraffe skyscrapers eating cloud leaves in whimsical metropolitan savanna today | 298 | Secluded monastery balanced atop waterfall edge within verdant hidden valley |

Table A1 – *Continued from previous page*

| # | Prompt | # | Prompt |
|---|--------|---|--------|
| 299 | Elegant cyborg ballerinas rehearsing in abandoned warehouse filled with mirrors | 300 | Thunderstorm brewing inside snow globe resting on antique wooden desk |

# B   Adaptation of Attend-and-Excite for SDXL

To employ Attend-and-Excite (A&E) as a baseline for the Stable Diffusion XL (SDXL) architecture, we adapted the official implementation to account for the model's increased latent resolution and dual text-encoder structure. While the core Generative Semantic Nursing (GSN) logic remains consistent with Chefer et al. (2023), our implementation introduces the following specific changes:

1. **Attention Map Extraction:** The original method operates on $16 \times 16$ attention maps. In SDXL, we found this resolution (corresponding to a 64-pixel stride) too coarse. We instead extract maps from the $32 \times 32$ cross-attention layers of the U-Net. We aggregate attention probabilities across heads and derive token indices using the tokenizer associated with the second text encoder (OpenCLIP ViT-bigG/14).

2. **Padding and Smoothing:** To prevent the optimization from exploiting edge artifacts, we apply *reflection padding* to the attention maps prior to Gaussian smoothing (kernel size $3 \times 3$, $\sigma = 0.5$). This ensures that subjects generated near the image boundaries are not penalized by zero-padding in the loss calculation. Additionally, we scale the raw attention scores by a factor of 100 before the Softmax operation to sharpen the attention distribution.

3. **Optimization Schedule:** We perform the latent optimization during the first 25 steps of a 30-step generation schedule. We utilize an iterative refinement strategy at timesteps $t \in \{0, 10, 20\}$ with acceptance thresholds $\tau \in \{0.1, 0.5, 0.8\}$. The latent code $z_t$ is updated with a gradient step size $\alpha_t$ that follows a linear decay schedule:

$$\alpha_t = S \cdot \left( 1 - 0.5 \cdot \frac{t}{T_{total}} \right) \tag{15}$$

where the initial scale factor $S$ is set to 20.0. Gradients are normalized if their norm exceeds $1 \times 10^{-5}$ to maintain stability within the SDXL latent space.

4. **Subject Selection:** We automate the selection of subject tokens using a Part-of-Speech (POS) tagger to identify nouns and proper nouns within the input prompts.

## C   Additional Qualitative Results

In this section, we provide further qualitative results to complement the main paper. These examples demonstrate our method's ability to improve prompt-image alignment and preserve visual fidelity across a diverse set of prompts.

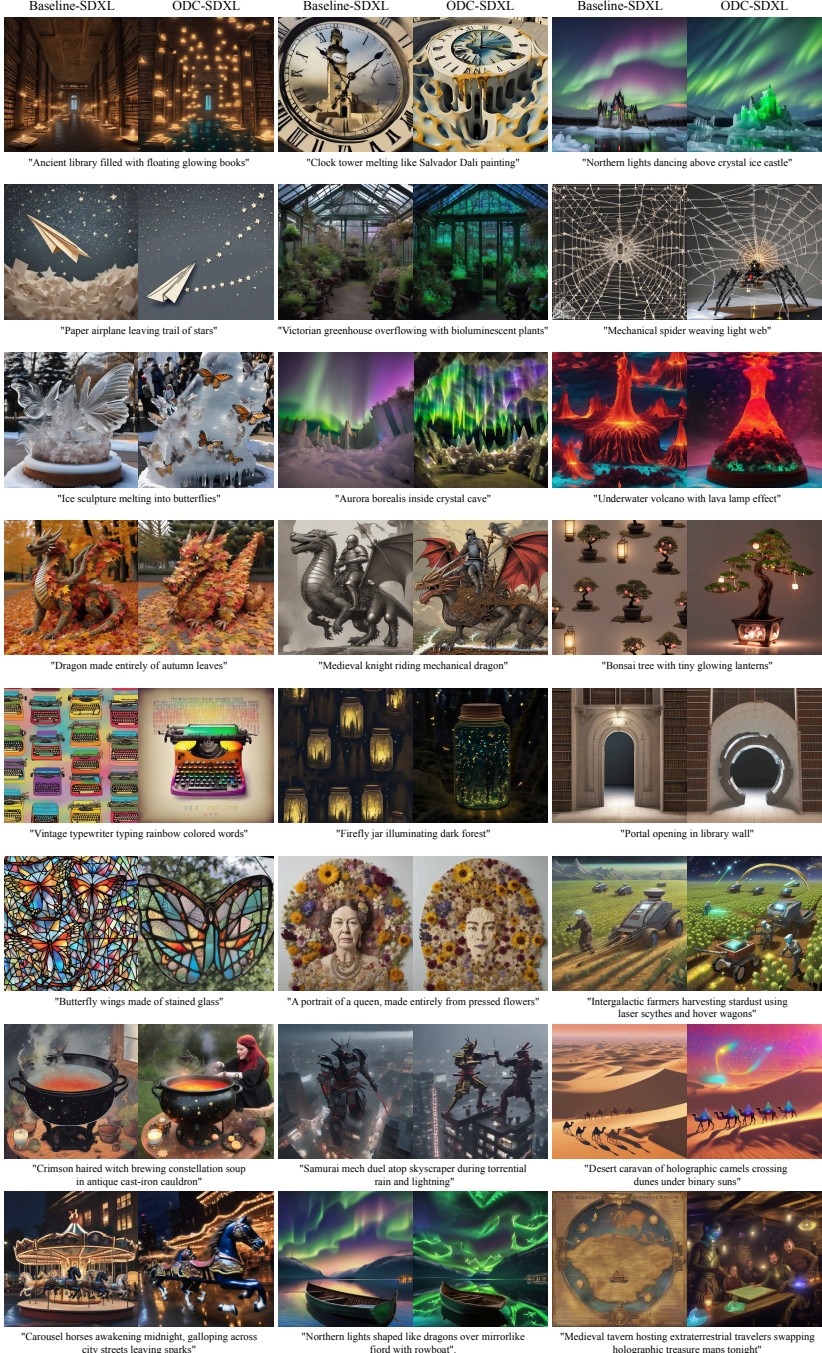

Figure A1: Additional qualitative results comparing baseline generations with Orthogonal Drift Correction (ODC). For each prompt, we show the output from the baseline SDXL model (left) and our ODC-corrected version (right).

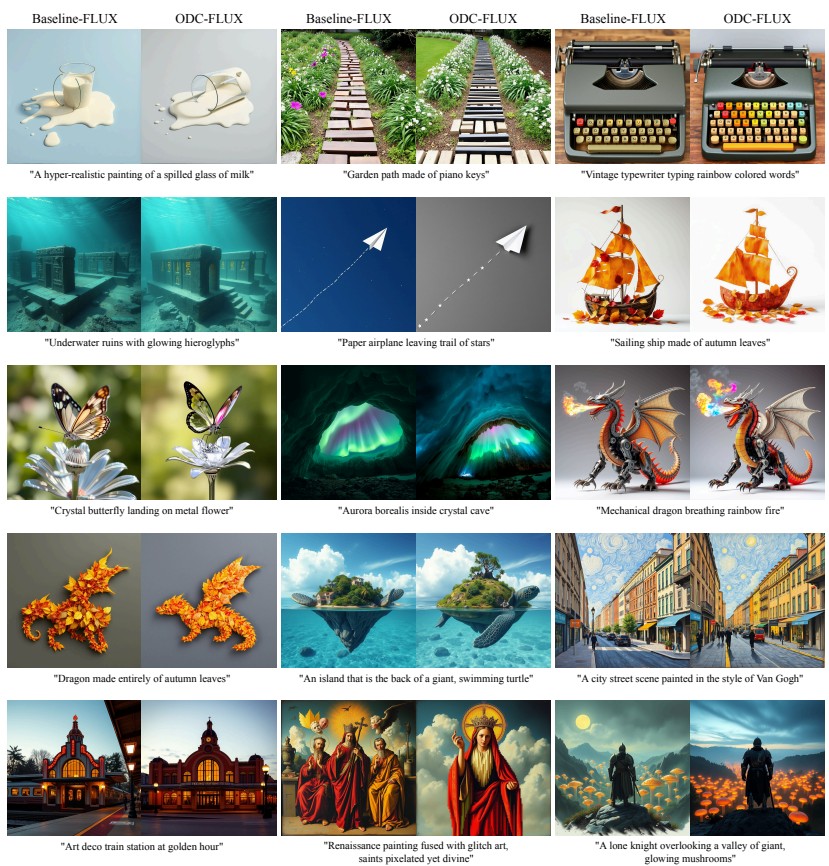

Figure A2: Additional qualitative results comparing baseline generations with Orthogonal Drift Correction (ODC). For each prompt, we show the output from the baseline FLUX model (left) and our ODC-corrected version (right).

