# OpenReview forum: "ODC: Orthogonal Drift Correction for Improved Text-to-Image Semantic Alignment at Inference"
_TMLR — Rejected by TMLR_

### Review · Reviewer_Urmr · 2026-04-19

**Summary Of Contributions:**

This paper proposes a training-free, inference-time guidance method to reduce semantic drift in text-to-image models, where outputs fail to align with complex prompts. The method, ODC, identifies misaligned (orthogonal) components in the text embedding by analyzing an initial generated image with a vision-language model, converts this error into discrete text via a large vocabulary retrieval dataset (300K+ terms), and removes it using an adaptive rank-reduced concept editing strategy. The authors also introduce COALBench-300, a benchmark of 300 compositional prompts for evaluating alignment. Experiments show that ODC, as a plug-and-play module, consistently improves alignment on models such as Stable Diffusion XL and FLUX.1 without requiring retraining or architectural changes.

The authors propose an interesting concept, and the qualitative results suggest that the method improves the model's ability to capture nuanced concepts and maintain prompt-image alignment in some cases. However, the current experimental evaluation leaves several gaps that hinder the paper's overall impact:
- Lack of human evaluation (user study): The core task of evaluating how well an image aligns with complex, creative intent is inherently subtle and subjective. A formal user study is necessary to validate the claims.
- Missing assessment of generation quality: The paper focuses heavily on alignment metrics but fails to quantitatively evaluate whether the base generation quality is degraded by the method. Improving text guidance should not come at the cost of overall visual fidelity.
- Absence of VLM ablation studies: The authors acknowledge in their limitations that ODC's effectiveness is strictly bound by the capabilities of the VLM used, yet the experiments rely only on BLIP2-ITM-ViT-G. An ablation study comparing different VLMs is necessary to demonstrate the system's sensitivity and robustness.
- Hyperparameter choices: The methodology introduces several parameters, such as retrieving the top k=5 items, using a softmax temperature of 0.5, and applying distinct concept removal weights of 1.0 for SDXL and 0.5 for FLUX. The paper does not provide ablation studies to justify these empirical selections or demonstrate how sensitive the outputs are to these values.
- Example of "error word" in qualitative results: Providing "error words" alongside the images would greatly improve the interpretability of the method.

**Audience:**

Yes

**Audience Explanation:**

The paper addresses an important problem within an active area of research. Although the proposed solution requires further evaluation to fully substantiate its claims, it demonstrates promising improvements in the qualitative results. Additionally, the introduction of a new evaluation dataset, COALBench-300, provides a useful resource for the community

**Broader Impact Concerns:**

There aren't broader concerns regarding the ethical implications of the work.

**Claims And Evidence:**

Yes

**Claims Explanation:**

The claims are partially supported. The paper does a solid job supporting their foundational mathematical claims, but their broader claims about generalizability and user experience lack the necessary rigorous testing.

**Requested Changes:**

The requested changes have been incorporated into the weaknesses section above.

---

### Review · Reviewer_jUy5 · 2026-04-26

**Summary Of Contributions:**

This paper proposes Orthogonal Drift Correction (ODC), a training-free inference-time method for improving prompt-image alignment in text-to-image diffusion models. ODC first generates an initial image, uses a vision-language model to estimate a semantic drift vector between the generated image and the prompt, isolates the component orthogonal to the prompt direction, retrieves corresponding terms from a large vocabulary of over 300K items, and uses the re-embedded drift representation in an adaptive concept removal module before a second generation pass. The authors also introduce CoalBench-300, a benchmark of 300 compositional prompts.

Strengths: The intervention point (refining text conditioning) is distinct from existing inference-time methods. The method is training-free and evaluated on two backbones (SDXL, FLUX).
Weaknesses: The method's effectiveness depends on the chosen VLM, with no analysis using alternative VLMs. No human evaluation. The presentation of the method section could be clearer.

**Audience:**

Yes

**Audience Explanation:**

Prompt-image alignment is an active research area in the T2I community, and training-free inference-time methods are of practical interest. The proposed intervention point (refining text conditioning before it enters the U-Net) is distinct from existing approaches that operate on cross-attention maps or latents, and the CoalBench-300 benchmark may be useful to researchers working on alignment and compositional generation.

**Broader Impact Concerns:**

No broader impact concerns. The work proposes an inference-time alignment correction method and does not introduce new generative capabilities or data resources requiring ethical discussion.

**Claims And Evidence:**

No

**Claims Explanation:**

While the paper presents empirical improvements over baselines, I find the evidence not fully convincing to support the strength of the claims.

First, the evaluation relies primarily on automatic metrics such as CLIPScore, BLIPScore, and benchmark-specific automatic scores. While these metrics are widely used and informative, they mainly serve as proxies for image–text alignment and may not fully reflect fine-grained compositional correctness. The absence of human evaluation makes it difficult to assess whether the reported improvements correspond to perceptually meaningful gains.

Second, the method depends critically on a vision-language model to estimate semantic drift, yet only a single VLM is used for this purpose. Without analyzing robustness to alternative VLMs, it remains unclear how generalizable the proposed approach is.

Third, while improvements are reported, in some settings they appear relatively modest, and there is limited analysis of their consistency or practical significance across prompts and models.

Overall, the results suggest that the method is promising, but additional validation would be needed for the claims to be fully convincing.

**Requested Changes:**

1. Critical: Provide an analysis of the method's robustness to the choice of VLM. Repeating the main experiments with one or two alternative VLMs (for example, SigLIP or EVA-CLIP, ideally chosen to be independent from the models used in the evaluation metrics) would clarify how much of the reported improvement depends on the specific BLIP2-ITM-ViT-G model.
2. Critical: Add a human evaluation, even on a subset of CoalBench-300 (for example, 30 to 50 prompts with pairwise comparisons against the vanilla baseline). Given the limitations of CLIP and BLIP scores on fine-grained compositional alignment, human judgments would substantially strengthen the main claims.

---

### Review · Reviewer_QXCd · 2026-04-28

**Summary Of Contributions:**

- The paper proposes Orthogonal Drift Correction (ODC), a training-free inference-time method for improving semantic alignment in text-to-image generation. The key idea is to first generate an initial image, estimate the semantic drift between the prompt and the generated image in a vision-language embedding space, isolate the drift component orthogonal to the prompt embedding, and then use a vocabulary-based surrogate together with an adaptive rank-reduced concept removal module to refine the text conditioning for a second generation pass.

- The paper also introduces CoalBench-300, a curated benchmark of 300 compositional prompts designed to stress-test prompt-image alignment. Experiments on SDXL and FLUX.1 [schnell], using CoalBench-300, PartiPrompts, and T2I-CompBench, show consistent improvements over vanilla generation and several inference-time baselines in CLIP/BLIP-based alignment metrics, with no additional memory cost but roughly doubled inference latency due to the second generation pass.

**Audience:**

Yes

**Audience Explanation:**

Yes. The paper addresses a relevant and widely recognized problem in text-to-image generation: improving semantic faithfulness to complex prompts without retraining the model. This is likely to interest researchers working on generative models, diffusion models, vision-language alignment, prompt-image evaluation, and inference-time control methods. The proposed intervention point—modifying the text conditioning embedding before generation rather than manipulating attention maps or optimizing latents during denoising—is a useful perspective that may inspire follow-up work. The new CoalBench-300 benchmark may also be of interest to researchers studying compositional prompt following.

**Broader Impact Concerns:**

The paper studies a method for improving semantic alignment in text-to-image generation. This can have positive impacts by reducing user frustration and improving the controllability of generative models. However, better prompt adherence can also make it easier to generate harmful, deceptive, or policy-violating visual content when the underlying generation model allows such content. The method is model-agnostic and training-free, so it could potentially be applied to models with weak safety filters.

I do not see a severe ethical concern that would prevent publication, but the paper should include a brief broader impact statement discussing the dual-use nature of improved prompt-image alignment, possible misuse for deceptive or harmful image generation, and the need to use ODC together with appropriate safety filtering and content moderation mechanisms.

**Claims And Evidence:**

Yes

**Claims Explanation:**

The main claims are generally supported by the evidence presented in the paper. The method is described clearly through equations, algorithms, and a workflow diagram, making the proposed two-stage correction process understandable and reproducible at the algorithmic level. The empirical evaluation covers multiple prompt sets and two different text-to-image backbones, SDXL and FLUX.1 [schnell], which provides evidence that the approach is not limited to a single model. The quantitative results show consistent gains in CLIPScore and BLIPScore on CoalBench-300 and PartiPrompts, and the comparison on T2I-CompBench further supports the claim that ODC improves compositional alignment. The qualitative examples also help illustrate the kinds of semantic drift that the method can correct.

That said, some claims would be more convincing with additional evidence. The improvements in BLIPScore are relatively small in several settings, and the paper relies heavily on automatic vision-language metrics that may not fully capture human-perceived prompt fidelity. A human evaluation or preference study would strengthen the evidence considerably. In addition, the ablation study mainly compares full-vector correction with orthogonal correction, but does not fully isolate the contributions of top-k selection, rank reduction, and the removal strength. The method also introduces approximately 2x latency, so the practical efficiency claim should be framed more carefully. Overall, however, the central claim that ODC can improve prompt-image alignment at inference time is supported.

**Requested Changes:**

- Add a human evaluation or preference study. Since the main goal is semantic alignment between text prompts and generated images, the paper should include human judgments comparing vanilla generation, ODC, and relevant baselines. Automatic metrics such as CLIPScore and BLIPScore are useful but can miss fine-grained compositional errors, attribute binding failures, and visual artifacts.

- Report statistical significance or confidence intervals. The improvements, especially for BLIPScore, are sometimes small. The authors should report standard deviations, confidence intervals, or significance tests across prompts/seeds to make clear whether the observed gains are robust.

- Strengthen the ablation study. The current ablation supports the use of the orthogonal component, but the method has several additional design choices. The paper should ablate the top-k value, removal strength α, rank-reduction bounds, and the use of adaptive rank reduction versus simpler concept removal.

- Provide more analysis of failure cases. The limitations section mentions dependence on the VLM and difficulty with abstract concepts, but the paper would benefit from concrete failure examples and an analysis of when ODC harms generation quality or prompt fidelity.

---

### Decision · Action_Editor_4Njy · 2026-06-14

**Recommendation:** Reject

**Additional Comments:**

The reviewers generally agreed that the problem is important and that the proposed inference-time correction mechanism is technically interesting. However, the evaluation is not yet sufficiently convincing for publication. In particular, human evaluation is essential for validating improvements in semantic alignment, and additional analyses are needed to demonstrate robustness and isolate the contributions of individual design choices. These concerns were consistently raised by multiple reviewers and remain unaddressed due to the absence of an author rebuttal.

**Audience:**

Yes

**Audience Explanation:**

The submission addresses the important problem of improving semantic alignment in text-to-image generation using a training-free inference-time method. The proposed conditioning refinement strategy represents a practically useful perspective distinct from many existing inference-time approaches, and the introduced benchmark may also be valuable to researchers studying compositional prompt following. The topic is likely to be of interest to researchers working on generative models, diffusion models, vision-language alignment, and inference-time control.

**Claims And Evidence:**

No

**Claims Explanation:**

The paper proposes a training-free inference-time approach for improving prompt-image alignment in text-to-image generation and presents encouraging empirical results across multiple benchmarks and two diffusion backbones. One reviewer considered the experimental evidence sufficient to support the main claims, citing consistent improvements and clear methodological presentation. However, two reviewers remained unconvinced that the evidence fully substantiates the paper's broader claims. The primary concerns are the lack of human evaluation, reliance on automatic alignment metrics, absence of robustness analysis across alternative vision-language models, limited ablations of key design choices, and insufficient analysis of generation quality, statistical significance, and failure cases. As the authors did not provide a rebuttal, these concerns remain unresolved. While the proposed method appears technically sound and promising, the current evidence is not yet sufficiently comprehensive to fully support the claimed effectiveness.

**Resubmission Of Major Revision:**

The authors may consider submitting a major revision at a later time.